# Impact of Suspected Preterm Labor during Pregnancy on Cardiometabolic Profile and Neurodevelopment during Childhood: A Prospective Cohort Study Protocol

**DOI:** 10.3390/diagnostics13061101

**Published:** 2023-03-14

**Authors:** Jesús González, Marina Vilella, Sonia Ruiz, Iris Iglesia, Marcos Clavero-Adell, Ariadna Ayerza-Casas, Angel Matute-Llorente, Daniel Oros, Jose Antonio Casajús, Victoria Pueyo, Gerardo Rodriguez, Cristina Paules

**Affiliations:** 1Pediatrics Department, Quirónsalud Hospital Zaragoza, 50006 Zaragoza, Spain; 2Instituto de Investigación Sanitaria Aragón (IIS Aragon), 50009 Zaragoza, Spain; 3Red RICORS “Primary Care Interventions to Prevent Maternal and Child Chronic Diseases of Perinatal and Developmental Origin”, RD21/0012/0001, Instituto de Salud Carlos III, 28029 Madrid, Spain; 4Growth, Exercise, Nutrition and Development (GENUD) Research Group, Instituto Agroalimentario de Aragon IA2 Universidad de Zaragoza, 50009 Zaragoza, Spain; 5Paediatric Cardiology Department, Miguel Servet University Hospital, 50009 Zaragoza, Spain; 6Centro de Investigación Biomédica en Red Cardiovascular (CIBERCV), Instituto de Salud Carlos III, 28029 Madrid, Spain; 7Department of Physiatry and Nursing, Faculty of Health and Sport Sciences (FCSD), University of Zaragoza, 22001 Huesca, Spain; 8Physiopathology of Obesity and Nutrition Networking Biomedical Research Center (CIBERObn), 28029 Madrid, Spain; 9Obstetrics Department, Hospital Clínico Universitario Lozano Blesa Zaragoza, University of Zaragoza, 50009 Zaragoza, Spain; 10Ophthalmology Department, Miguel Servet University Hospital, University of Zaragoza, 50009 Zaragoza, Spain; 11Pediatrics Department, Hospital Clínico Universitario Lozano Blesa, University of Zaragoza, 50009 Zaragoza, Spain

**Keywords:** suspected preterm labor, preterm birth, fetal programming, neurodevelopment, cardiometabolic profile

## Abstract

Introduction: Suspected preterm labor (SPL), defined as the presence of regular and painful uterine contractions and cervical shortening, represents a prenatal insult with potential long-term consequences. However, despite recent evidence demonstrating suboptimal neurodevelopment at 2 years in this population, it remains underestimated as a significant risk factor for neurodevelopmental disorders or other chronic diseases. The aim of this study is to assess the impact of suspected preterm labor during pregnancy on cardiometabolic profile and neurodevelopment during childhood (6–8 years). Methods and analysis: Prospective cohort study including children whose mothers suffered suspected preterm labour during pregnancy and paired controls. Neurodevelopmental, cardiovascular, and metabolic assessments will be performed at 6–8 years of age. A trained psychologist will carry out the neurodevelopment assessment including intelligence, visual perception, and behavioral assessment. Body composition and physical fitness assessment will be performed by one trained pediatrician and nurse. Finally, cardiovascular evaluation, including echocardiography and blood pressure, will be performed by two pediatric cardiologists. Data regarding perinatal and postnatal characteristics, diet, lifestyle, and weekly screen time of the child will be obtained from medical history and direct interviews with families. Primary outcome measures will include body mass index and adiposity, percentage of fat mass and total and regional lean mass, bone mineral content and density, cardiorespiratory resistance, isometric muscle strength, dynamic lower body strength, systolic and diastolic blood pressure, left ventricle (LV) systolic and diastolic function, general intelligence index, visuospatial working memory span, oculomotor control test, index of emotional, and behavioral problems.

## 1. Introduction

Fetal programming has been defined as the intrauterine environment at critical stages of the pregnancy which are decisive for the long-term development of the fetus and child [1,2]. Suspected preterm labour (SPL), defined as the presence of regular and painful uterine contractions and cervical shortening [3], represents a prenatal insult with potential long-term consequences. This condition occurs in approximately 9% of all pregnancies, being the leading cause of hospital admission during pregnancy, excluding labor at term. Within this population of women, the incidence of preterm birth is 30–40%. It is well known that preterm birth accounts for approximately 70% of neonatal morbidity and mortality [4] and is a risk factor for the development of neurological and cognitive deficits [5,6] and impairment of body composition, metabolism, and cardiovascular system [7]. However, there is a widespread belief that there are no long-term effects for the fetus, neonate, or infant if birth occurs at term, considering the episode of preterm labor benign and sometimes labeling it as “false preterm labor” [8]. Increasing evidence reported in recent years suggests that this conclusion may not be correct and further studies with long-term follow-up are necessary [9].

### 1.1. Suspected Preterm Labour and Neurodevelopment during Childhood

Prematurity has been associated with neurodevelopmental deficits that warrant special educational services [10,11,12]. Although healthy late-preterm infants seem to have greater risks of developmental delay, disability, and school-related problems throughout childhood [13,14,15], the impact of late preterm birth on cognition is not fully characterized. Methodological issues, such as correcting for age at prematurity, may explain conflicting results between different studies [12,16,17].

It is well known that visual impairment is more common in preterm infants, and its incidence is inversely proportional to body weight and gestational age at birth, with a higher incidence in early preterm infants [18]. Nevertheless, disturbances in visual development have also been observed in late preterm infants, demonstrating lower visual acuity and contrast sensitivity, in addition to higher rates of strabismus [19].

Moreover, preterm infants are at higher risk for social-emotional and academic performance problems, compared to full-term infants [20]. In fact, higher rates of these problems have been reported in children born preterm compared to other more serious neurodevelopmental problems, such as cerebral palsy, intellectual disability, epilepsy, and severe visual/hearing disorders [19,21,22].

Although neurobehavioral competencies are mainly associated with neurological maturation [23,24] preliminary evidence suggests that adverse outcomes in infants born late preterm or early term may not only be due to physiological immaturity but also to other biological determinants [25]. Consistent with this hypothesis, our group reported that an episode of suspected preterm labour is a risk factor for neurodevelopmental deficits at two years of age. Furthermore, the pattern of developmental deficits in children born at term after threatened preterm labor was similar to that of children born late preterm [8]. Therefore, a more exhaustive follow-up of these children later in childhood is important for better characterization.

### 1.2. Suspected Preterm Labour and Cardiometabolic Profile during Childhood

Neonatologists and pediatricians wish a progressive and adequate growth within the same percentile; however, it is usual for premature patients to suffer growth restrictions and accelerations, as a consequence of some pathology associated with their prematurity or due to their condition as premature babies. Thus, it has been described those premature patients, once reaching their full-term age, present an increase in fat mass and fat mass index, but not in lean mass, compared to controls born at term [26,27,28].

Prematurity also causes functional limitations, such as decreased lung function [29] or cardiorespiratory capacity [30]. Increased blood pressure, signs of increased peripheral vascular resistance, and cardiac remodeling can be observed in early childhood in preterm-born children [31,32,33,34]. Moreover, preterm-born children have reduced insulin sensitivity and a higher risk for markers of metabolic syndrome when there is excessive childhood weight gain [35,36]. Nevertheless, research has been focused on very early preterm infants with few or no studies in late preterm infants or in those born at term after suffering preterm labor during pregnancy.

### 1.3. Justification for the Study

The high incidence of suspected preterm labor (SPL) and its potential impact on multiple organ functional remodeling during critical developmental periods, requires more comprehensive research on the long-term consequences in these children. Despite evidence linking suspected preterm labor and late prematurity to suboptimal health outcomes, they remain under-recognized as a significant risk factor for neurodevelopmental disorders or other chronic diseases. Further investigations are essential to properly define health outcomes in childhood and develop strategies to improve the global health of this population.

## 2. Objectives

Primary objective:

To assess the impact of SPL during pregnancy on cardiometabolic profile and neurodevelopment during childhood (6–8 years).

Secondary objectives:Describe the prevalence of obesity and the different patterns of body composition and bone mineral density in childhood associated with SPL.Determine the systolic and diastolic blood pressure of children born after suffering an SPL during pregnancy.Evaluate the cardiorespiratory resistance and muscular strength in children born after suffering an SPL during pregnancy.Describe the different patterns of cardiovascular remodeling in childhood associated with SPL.Assess global intelligence and visual perception in children born after suffering an SPL during pregnancy.Evaluate clinical and adaptive behavior of children born after suffering an SPL during pregnancy.

## 3. Methods and Analysis

### 3.1. Study Design

Prospective cohort study including children whose mothers suffered a suspected preterm labor during pregnancy and paired controls. Pregnant women were recruited during pregnancy and children have been followed up into childhood. At 2 years of age, a neurodevelopmental study was performed on all these children [9]. In the current study, we carry out a neurodevelopment, cardiovascular, and metabolic assessment at 6–8 years of age (Figure 1).

Suspected preterm labor was defined [3] as the presence of regular and painful uterine contractions documented by cardiotocography and ultrasound cervical length less than 25 mm [37] in the presence of intact membranes at gestational age ranging from 24 + 0 to 36 + 6 weeks. Pregnancies were dated according to first-trimester crown-rump length [38].

Tocolysis with atosiban (Tractocile, Ferring Pharmaceuticals, Madrid, Spain) and intramuscular betamethasone (12 mg/24 h, 2 doses) were performed for some cases according to manufacturer recommendations and international clinical standards [39].

After families agree to participate in the study, data regarding perinatal and postnatal characteristics, diet, lifestyle, and weekly screen time of the child will be collected from clinical histories and direct interviews with the families. We will perform three different assessments on two different days. The first one, a trained psychologist will carry out the neurodevelopment assessment in the afternoon in a quiet room. After that, on the same day, body composition and physical fitness assessment will be performed by one trained pediatrician and nurse. Finally, at the same time, direct interviews with the families will take place. On the second day, only cardiovascular assessment will be performed by two pediatric cardiologists who had previously been trained for consistency between measurements.

All the researchers involved in the clinical protocol will be blinded to group and perinatal outcomes.

### 3.2. Participants

Two cohorts of children will be assessed at 6–8 years of age:(a)Exposed cohort: children whose mothers suffered a suspected preterm labor during pregnancy. They are subdivided into:
∘Moderate-late preterm births (between 32 and 37 weeks of gestation)∘False suspected preterm labor (full-term births (≥37 weeks) after suffering suspected preterm labor)(b)Unexposed cohort: children born at term (≥37 weeks) whose mothers did not suffer suspected preterm labor during pregnancy.
Inclusion criteria: singleton pregnancy, good understanding, and signed informed consent.Exclusion criteria: multiple pregnancy, malformations, chromosomal diseases, or prenatal infections.

### 3.3. Setting

This study includes children born at the Hospital Clínico Universitario Lozano Blesa (Zaragoza, Spain) between 2011 and 2013. Neurodevelopment and metabolic assessment will be performed at the University of Zaragoza. Cardiovascular assessment will be performed at the Pediatrician Cardiology Unit of Hospital Miguel Servet (Zaragoza, Spain).

### 3.4. Patient Involvement

Patients were involved in the design and conduct of this research. During the feasibility stage, the priority of the research question, choice of outcome measures, and methods of recruitment were informed by discussions with patients. Once the study will be published, participants will be informed of the results and will be sent details of the results.

### 3.5. Primary Outcome Measures

The main outcome variables are body mass index and adiposity, percentage of fat mass and total and regional lean mass, bone mineral content and density, cardiorespiratory resistance, isometric muscle strength, dynamic lower body strength, systolic and diastolic blood pressure, left ventricle (LV) systolic and diastolic function, general intelligence index, oculomotor control (fixation stability, duration of fixations, saccadic reaction time, and saccadic precision), visual discrimination, visual memory, spatial relationships, form constancy, sequential memory, visual figure ground and visual closure, the global index of problems, index of emotional problems, index of behavioral problems, index of problems in executive functions, and index of resources personal.

### 3.6. Covariables

Multiple underlying factors could potentially affect neurodevelopment, body composition, or cardiovascular health, so the following covariates will be actively evaluated:Baseline and socioeconomic characteristics: age, ethnicity, clinical characteristics, socioeconomic level, and educational level of the parents.Perinatal characteristics: parity (number of deliveries > 22 weeks), assisted reproductive technology, history of preterm delivery, pregnancy outcomes (pre-eclampsia, gestational diabetes, small for gestational age), administration of prenatal corticosteroids(gestational age at treatment and number of doses), administration of tocolytics during the pregnancy, type of delivery, birthweight (birthweight will be adjusted by the mother’s BMI at delivery), Apgar test, umbilical pH, NICU admission, and neonatal outcomes (bronchopulmonary dysplasia, necrotizing enterocolitis, sepsis, intraventricular hemorrhage, periventricular leukomalacia, retinopathy, patent ductus arteriosus).Lifestyle of the mother during pregnancy: smoking, alcohol consumption, or other drugs. Body mass index at the beginning and end of gestation.Feeding in the first two years of the child’s life: breastfeeding, age of introduction of complementary feeding.Body composition of the newborn and the child. The variables will be obtained in each child by the nursing and pediatric staff of the Primary Care centers during routine visits.Diet and lifestyle of the child: Dietary intake will be self-reported by parents through a semi-quantitative food frequency questionnaire (FFQ) [40,41] that has been previously used and validated in the multifactorial evidence-based approach using behavioral models in understanding and promoting fun, healthy food, play, and policy for the prevention of obesity in early childhood study (ToyBox-study). In short, the FFQ consists of a list of foods and beverages with response categories to indicate the usual frequency of consumption over the selected period. We will calculate the Diet Quality Index (DQI), which is a largely used index in cohorts with similar characteristics, in order to assess diet in terms of three subcomponents: dietary diversity, quality, and equilibrium [42]. Physical activity and a sedentary lifestyle will be also self-reported by parents using IPAQ-C questionnaires [43].Weekly screen time (WST): Using a previously validated questionnaire, parents reported the number of hours of TV/DVD/video viewing and computer/game console use of their child both for a typical day on weekdays and on weekend days. We summed the reported hours per day on weekdays and weekend days to obtain the total WST (hours on weekdays + hours on weekend days/7 days per week) [43].Intercurrent processes: Infections, admissions to hospital, surgical interventions, traumatic processes.

### 3.7. Children and Parents Assessment

We will perform a neurodevelopment, cardiovascular, and metabolic assessment at 6–8 years of age, evaluating:

(a)Body Composition
Height will be measured with a stadiometer with a precision of 0.1 cm (SECA 225, SECA, Hamburg, Germany), and weight with a scale precision of 0.1 kg (SECA 861, SECA, Hamburg, Germany). Determination of z-score values of BMI for age (z-BMI) for girls and boys will be performed using the WHO Anthro Software, according to the WHO growth standards of 2006–2007 [44].BMI will be adjusted by the mother´s and father’s BMI at the time of assessment.Fat mass, lean mass, and bone mass will be determined by dual-energy X-ray absorptiometry (DXA). DXA scans will be performed in a supine position, wearing light clothing with no metal and no shoes or jewelry. All DXA scan tests will be analyzed by the same researcher using a Hologic Horizon A scanner and a pediatric version of the software APEX, Hologic Corp., software version 5.5.1.2 (Bedford, MA, USA). From the regional and total analysis of the complete body scan, the following will be assessed: total fat mass, subtotal (total without head), trunk, and limbs. The total area (cm^2^) and bone mineral content (CMO; g) will be calculated. Bone mineral density (BMD; g/cm^2^) will be calculated following the formula BMD = CMO(Area-1). Two additional analyzes will be carried out to estimate bone mass at the level of the lumbar spine and the left hip [45]. Lean mass (body mass– (FM + bone mass)), percentage body fat mass (percentage of fat grams/total mass). Fat mass index (FMI) will be a continuous variable calculated for each participant from data obtained from DXA as a fat mass in kilograms/height in square meters. Additionally, fat-free mass index (FFMI) kg/height in meters^2^ will be also used in this study [46]. Abdominal adiposity will be assessed at a delimited region that will be drawn on the digital scan image, delimiting the lower horizontal border on the top of the iliac crest and the upper border parallel to the end of the lowest rib [47].(b)Physical Fitness Assessment
Cardiorespiratory FitnessBefore starting data collection, the participants were familiarized with the laboratory and procedures. After fitting the safety harness, the test began when the participants were able to walk easily on the treadmill (Quasar Med 4.0, h/p/cosmos, Nußdorf, Germany). The protocol started (depending on height) with a speed of 2.4 km/h or 3.2 km/h, increasing by 0.8 km/h every 2 min until the participants were unable to walk or reached a speed of 4.8 km/h or 5.6 km/h. Then, the slope was increased by 4% every minute until exhaustion or up to a maximal slope of 24%. A sports medicine physician supervised the entire test and performed a pre-clinical examination to determine if the participant was suitable for performing the stress test. The respiratory gas exchange data were measured breath-by-breath using open-circuit spirometry (Oxycon Pro, Jaeger/Viasys Healthcare, Hoechberg, Germany). Peak oxygen uptake (VO_2peak_) values were averaged over consecutive 15 s periods. The metabolic cart’s daily calibration was performed with a known gas and volume as recommended by the manufacturer. Heart rate (HR) was continuously recorded using 12-lead electrocardiography (H12+, Mortara Instrument, Milwaukee, WI, USA) from the beginning to the end of the stress test. Low and high cardiorespiratory categories were established using the 50th percentile by sex published by Johansson et al. [48] for overweight and obese child populations. The reference 50th percentile values for relative VO_2max_ in boys and girls were 30.8 and 30.6 mL/kg/min, respectively. This test has been carried out in several research projects in a population of the same age with and without intellectual disabilities for approximately 10 years [49].Handgrip strength tests (handgrip):This will be measured using a handgrip dynamometer (TKK 5001, grip A; Takei) (range, 0–100 kg; accuracy, 0.5 kg). Children will be in a standing position maintaining the arm of the tested side straight down with the shoulder slightly abducted (~10° not touching the rest of the body), the elbow in 0° of flexion, the forearm in a neutral position, and the wrist in 0° of flexion. The best value of 3 attempts will be chosen [50].Lower limb explosive strength (SLJ):This test will consist of jumping horizontally, with both feet at the same time, the maximum distance over a non-slippery and hard surface with the feet immediately behind the starting line and separate from each other, approximately at shoulder width. Children will perform 3 jumps with 1 to 2 min of rest between attempts. The best value of 3 attempts (in centimeters) will be used for analysis [50].
(c)Cardiovascular Assessment
Blood pressureSystolic and diastolic blood pressure will be measured (OMRON M6, HEM 70001; Omron, Kyoto, Japan) with participants seated in a separate quiet room for 10 min with their backs supported and feet on the ground. Two readings will be taken at 10-min intervals, and the lowest measure will be recorded.EchocardiographyEchocardiographic examinations will be performed using a Siemens Acuson SC2000 ultrasound system (Siemens Healthcare, Erlangen, Germany) equipped with a 4 MHz sector transducer. Two pediatric cardiologists will perform all the echocardiographic examinations, following current guidelines. Image acquisition procedures will be harmonized before the study started. The optimal frame rate will be used to optimize myocardial deformation analysis. ECG-guided, we systematically will record three 2D cardiac-cycle loops in the following views: long-axis parasternal view, four-chamber apical view, three-chamber apical view, and two-chamber apical view. We also will use pulse-wave Doppler, tissue Doppler, M-mode, and speckle tracking to analyze the cardiac function and heart flows. The following LV measurements and function parameters will be studied: end-diastolic and end-systolic diameter, septum and posterior wall thickness, heart rate, ejection fraction (by Teicholz and biplane Simpson methods), mitral E and A wave’s velocity (pulse-wave Doppler), mitral annulus S´, E´ and A´ waves velocity (tissue Doppler), TEI index (tissue doppler), and LV strain and strain rate at three apical views (speckle tracking).
(d)Intellectual Assessment
Reynolds Intellectual Screening Test (RIST)General cognitive aptitude will be measured using RIST (Reynolds Intellectual Screening Test). This test is composed of two tasks: the Guess What (GW) subtest for the verbal score, where three pieces of information are given and the child has to guess the target object or idea, and the Odd Item Out (OIO) subtest for the non-verbal score, where several pictures are shown, and the child has to point to the incongruent one. These two tasks will provide an intelligence quotient (IQ) estimation for children and adults between 3 and 94 years of age [51].Visual Perception AssessmentTest of Visual Perceptual Skills—Third Edition (TVPS-3)TVPS is an individually administered, standardized test that assesses non-motor visual-perceptual skills in populations aged 5 to 21. With a multiple-choice response structure, 112 black-and-white visual stimuli on test plates will be presented and participants will indicate their response verbally or through gestures. The raw score will be obtained for each subtest by adding together the number of correctly answered items. Raw scores are then converted to standardized scores. We will obtain seven domains of visual perception: visual discrimination, visual memory, spatial relationships, form constancy, sequential memory, visual figure-ground, and visual closure [52].Oculomotor controlOculomotor control will be assessed by DIVE (DIVE Medical SL, Zaragoza, Spain). This digital tool includes a 12-inch screen of 2160 × 1440 pixels, corresponding to a visual angle of 22.11 degrees horizontally and 14.81 degrees vertically, and an eye tracker located below the screen to record all eye movements during the test. The maximum temporal resolution of the eye tracker is 120 Hz.DIVE includes carefully designed and validated visual stimuli to assess oculomotor control in children; fixation and saccadic performance will be recorded, binocularly and monocularly. The exam will be performed in a quiet room under mesopic ambient illumination. Children will be positioned on a chair approximately 65 cm from the screen and asked to fixate on the different stimuli on the screen, trying not to move their heads.A calibration procedure of the eye tracker will be executed before the fixation study. The fixation task involves two parts. The first part presents a long fixational task. It consists of a high-contrast cartoon of a child of 3 degrees (deg) × 1.56 deg appearing on the center of the screen, who will talk to the child for 10 s. The second part presents a short fixational task. The fixation target consists of pictures of a balloon, a bee, or a cat. Eight different visual stimuli will be randomly displayed all over the screen with a fixed distance of 9.26 deg between every two consecutive stimuli. Each stimulus will be presented for three seconds, with no stimuli overlapping.The saccadic performance will be assessed by the saccadic reaction time (SRT), defined as the lapse of time between the presentation of the new stimulus and the initiation of the saccadic movement towards the stimulus.We will calculate fixation stability by the bivariate contour ellipse area (BCEA), which quantifies the area in square degrees of the ellipse containing a certain percentage of the fixation positions registered during the measurement procedure. Consequently, a smaller value for BCEA is indicative of greater fixation stability [53].(e)Behavioral Assessment

Children and Adolescents Assessment System (SENA, from its Spanish acronym)

SENA is a multimethod, multidimensional system used to evaluate the behavior of children and young adults ages 3 through 18 years old. Three separate rating forms comprise the SENA. These include a teacher rating scale (TRS), a parent rating scale (PRS), and a self-report of personality form (SRP). We will use the parent rating scale as a comprehensive measure of a child’s adaptive and behavioral problems in community and home settings. The PRS uses a five-choice frequency response format; it is composed of 129 items and it takes 10 to 20 min to complete. The parent rating scale has three age-level forms: preschool, child, and adolescent.

The raw scores are manually entered into the Q-global software, which provides three block scales: problems block scale, vulnerability block scale, and personal resources block scale. From the combination of these scales, we will be able to obtain five indexes: global index of problems, index of emotional problems, index of behavioral problems, index of problems in executive functions, and index of personal resources [54].

All tests will be performed in a quiet room to avoid distractions and will be conducted by a professional clinical psychologist, trained in neuropsychological test administration.

### 3.8. Sample Size and Statistical Analysis

The sample size is defined considering the main objective of the study and the magnitude of the change in the main variable. In this project, different variables are relevant in this regard; therefore, a simulation of the sample size calculation will be carried out in reference to three variables of interest: total fat mass index, VO2max, and intelligence test score.

For type I error, α (the probability of incorrectly rejecting the null hypothesis),

We will assume a value of 0.05. For type II error, β (the probability of incorrectly accepting the alternative hypothesis), we will assume a frequent value of 20%; β = 0.2, therefore a statistical power of 0.8.

To perform the calculations, the G*Power tool (version 3.1.9.2 for Mac) was used, applying the sample size test for Student’s t-tests for two independent samples, since it was intended to compare differences between groups.

To perform the calculation, we will use the values of α and β, previously defined, and of the effect size, in this case, Cohen’s d value for each of the variables of the study. Reviewing previous studies, it is observed that the differences in the variables reach effect sizes of 0.6 for the total fat mass index [55], 0.83 for VO2max [56], and 0.67 for the score of the intelligence test [57].

Taking as a reference for the study the highest value of the three variables, the total size of the sample would be N = 90, with 45 participants in each group. However, the number of subjects to be recruited also depends on the possible losses: N’ = N/(1 − p), so if we consider the percentage of losses of 20%, the number of participants to be recruited would be N’ = 112.

As a general rule, qualitative variables will be described as absolute frequencies and relative percentages, quantitative variables as means and medians for the assessment of central tendency, and SD and IQR for the assessment of dispersion. In the case of ordinal variables, the description of both forms will be evaluated. Univariate analysis: for the comparison of two qualitative variables, the χ^2^ test or Fisher’s exact test will be used. When the variables are quantitative, the Student’s t-test for independent samples or the Mann—Whitney U test will be used if the applicability criteria are not met. Multivariate analysis will be performed by means of multiple (continuous variables) or logistic linear regression (categorical variables) controlling the possible confounding factors. For statistical analysis, values of *p* ≤ 0.05 will be considered statistically significant. The data will be analyzed with the SPSS program (V.20.0, IMB or newer).

## 4. Expected Results from This Study

In a previous study, we demonstrated that an episode of suspected preterm labor was a risk factor for neurodevelopmental deficits at 2 years of age. Compared with children born at term without suspected preterm labor, children born at term after suspected preterm labor scored significantly lower in all cognitive domains except for the expressive language. Moreover, the pattern of developmental deficits in children born at term after suspected preterm labor was similar to that of children born late preterm. These findings challenged the concept of “false preterm labor” is always a benign condition.

Consistent with this hypothesis, we designed the current study expecting to demonstrate that an episode of suspected preterm labor during pregnancy determines health consequences in childhood. Specifically, we hypothesize that an episode of suspected preterm labor leads to a higher percentage of obesity and greater adiposity during childhood and associates changes in cardio-respiratory resistance, blood pressure as well as in muscular strength. Furthermore, we expect to confirm that the neurodevelopmental deficits demonstrated previously persist during childhood.

### Ethics and Dissemination 

The study will follow the ethical standards of the Nuremberg Code of 1947 and the Declaration of Helsinki of 1964 as revised by Fortaleza (2013). The confidentiality and anonymity of the study participants will be guaranteed. In the case of detecting any abnormal result in any of the evaluated variables, the participants will be informed and, if necessary, will be assessed by the pediatric specialist.

Participant information sheets and informed consent have been approved by the research ethics committee (C.P.- C.I. PI19/392) and they will be delivered and explained to each of the participants. Study outcomes will be disseminated at international conferences and published in peer-reviewed scientific journals. Lay reports will be made available to study participants on request.

## Figures and Tables

**Figure 1 diagnostics-13-01101-f001:**
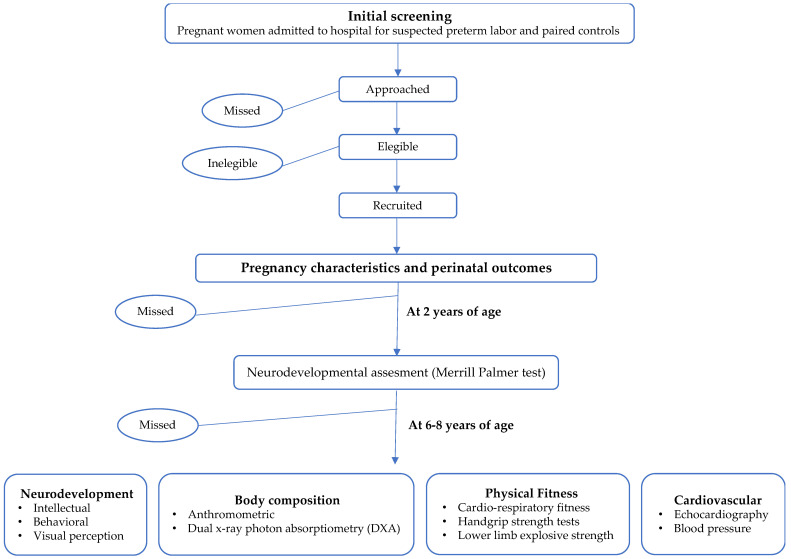
Schematic overview of study recruitment and follow-up.

## Data Availability

Not applicable.

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
