# Peer review of "Impact of Suspected Preterm Labor during Pregnancy on Cardiometabolic Profile and Neurodevelopment during Childhood: A Prospective Cohort Study Protocol"

_diagnostics, 2023, doi:10.3390/diagnostics13061101_

Round 1

Reviewer 1 Report

-Do you have any concerns regarding the effect of atosiban on your findings (neurobehaviour for example)? Are any other tocolytics used in your country (eg, nifedipine)? If yes, I think it might be a good idea to compare children that have received atosiban to those that have received a different tocolytic as for the primary outcome.

-Bearing in mind the possible detrimental effect that corticosteroids might have on different brain functions, I think to would be useful to to group your results according to the gestational week that steroids were given.

Reviewer 2 Report

Gonzales et al intend to study cardiometabolism and neurodevelopment in  preterm children at age 6-8 years. The protocol is written in good english and established with evidenced  based literature. My only concern is that preterm children have visual and respiratory difficulties.

I would suggest that relevant examinations by expertised would much decipher to the study goal they have.

Some amendements would much improve the protocol goals:

1. BMI adjustment of mothers during labor
https://doi.org/10.1093/ije/dyv151

2. BMI adjustment of children at age 6-8 years (this could be a second sub-study in the same study)

3. study of cardiometabolism without respiratory examination is incomplete, and strongly suggested for children been born preterm
https://doi.org/10.1586/ers.10.59

Eye examination starting from age 1 year in preterm babies is suggested in literature: i.e http://dx.doi.org/10.1136/bjo.84.9.963, https://doi.org/10.1038/s41433-022-02207-y, doi: 10.3389/fped.2022.819998.

Yet, I acknowledge that this latter (eye examination) might be a future study goal.

Round 2

Reviewer 2 Report

Most of my concerns have been addressed. I understand the limitations due to cost. Still, minor comments have to be addressed: 

a. Language editing (minor changes though)

b. In  Figure 1 there are 2 flow charts. Please name them a and b. on plot as well as in captions and text, detailing each one separately. Additionally, in every box of the chart add the number of subjects they refer if they have the information.

Author Response

Reviewer 2: 

Most of my concerns have been addressed. I understand the limitations due to cost. Still, minor comments have to be addressed: 

  1. Language editing (minor changes though)

Thank you for your suggestion. We had sent the manuscript to language editor in the previous review but we have sent again to improve it.  

  1. In  Figure 1 there are 2 flow charts. Please name them a and b. on plot as well as in captions and text, detailing each one separately. Additionally, in every box of the chart add the number of subjects they refer if they have the information.

The reviewer is right. We have merged the two flow charts into one graph. However, we decided not to include the number of subjects as we were unable to complete all the boxes of the chart at this point in the study.
